# Nucleosome disassembly during human non-homologous end joining followed by concerted HIRA- and CAF-1-dependent reassembly

**Xuan Li, Jessica K Tyler\***

Department of Pathology and Laboratory Medicine, Weill Cornell Medicine, New York, United States

**Abstract** The cell achieves DNA double-strand break (DSB) repair in the context of chromatin structure. However, the mechanisms used to expose DSBs to the repair machinery and to restore the chromatin organization after repair remain elusive. Here we show that induction of a DSB in human cells causes local nucleosome disassembly, apparently independently from DNA end resection. This efficient removal of histone H3 from the genome during non-homologous end joining was promoted by both ATM and the ATP-dependent nucleosome remodeler INO80. Chromatin reassembly during DSB repair was dependent on the HIRA histone chaperone that is specific to the replication-independent histone variant H3.3 and on CAF-1 that is specific to the replication-dependent canonical histones H3.1/H3.2. Our data suggest that the epigenetic information is re-established after DSB repair by the concerted and interdependent action of replication-independent and replication-dependent chromatin assembly pathways.

**\*For correspondence:** jet2021@
med.cornell.edu

**Competing interest:** See
page 17

**Reviewing editor:** Jerry L
Workman, Stowers Institute for
Medical Research, United States

## Introduction

Decades of studies have emphasized the critical importance of chromatin components, whose nature and spatial organization regulate cellular function and identity, including DNA repair (*Deem et al., 2012*). DNA double-strand breaks (DSBs) occur intrinsically during normal cell metabolism, or are caused by exogenous agents, such as ionizing radiation (IR) or some classes of chemotherapeutic drugs. In addition, DSBs are essential intermediates during programmed recombination events, such as meiosis, budding yeast mating-type switching, and lymphocyte development (*Mimitou and Symington, 2011*). Unrepaired and/or inappropriately repaired DSBs can be cytotoxic lesions, posing a severe challenge to maintaining genome integrity. Eukaryotic cells have evolved two mechanistically distinct pathways to repair DSBs: non-homologous end joining (NHEJ) and homologous recombination (HR) (*Ceccaldi et al., 2016*). In mammalian cells NHEJ is responsible for repairing the majority of DSBs throughout the cell cycle, while HR is mostly active in late S and G2 phases when a sister homolog is present (*Kass and Jasin, 2010*). Currently a major challenge is to understand how the dynamic states of chromatin influence these repair mechanisms.

Within the cell, eukaryotic DNA is packaged into the nucleoprotein structure known as chromatin. The nucleosome is the primary level of chromatin structure, with about 146 bp of DNA wrapped around an octamer of histones, which includes two molecules each of histones H2A, H2B, H3 and H4 (*van Holde, 1989*). Due to the intimate nature of the histone-DNA interactions within the nucleosome, the local chromatin structure has to be disrupted to allow the cellular machinery to gain access to DSBs to mediate their repair. How the chromatin changes to allow DSB repair to happen is

not clear. Also, how the chromatin structure, and the epigenetic information that it carries, is reestablished after DSB repair is complete, is poorly understood.

Upon generation of DSBs, the local chromatin structure is altered at many different levels, including histone post-translational modifications, chromatin relaxation, histone exchange, histone mobilization and histone removal from the DNA (*Deem et al., 2012*). The last of these processes, chromatin disassembly, ultimately allows free access of the repair machinery to the DNA lesion. Complete nucleosome disassembly, indicated by the removal of histone H3 from the DNA, has been observed in the vicinity of an unrepairable DSB undergoing extensive resection in budding yeast (*Tsukuda et al., 2005*). This chromatin disassembly was dependent on MRE11 and the INO80 ATP-dependent nucleosome remodeling complex (*Tsukuda et al., 2005*), both of which are required, either directly or indirectly, for timely DNA resection (*Paull and Gellert, 1998*; *van Attikum et al., 2004*; *Gospodinov et al., 2011*). Similarly, during HR in yeast, at least some nucleosomes are disassembled from around DSBs, because inhibition of the chromatin reassembly process reduced the local histone occupancy after DSB repair (*Chen et al., 2008*). These yeast studies provide precedent for local histone removal from around a DSB, seemingly dependent on DNA end-resection, and indicate that histone reassembly occurs during DSB repair to reestablish the genomic packaging and potentially the epigenetic information.

Chromatin disassembly and reassembly also appear to occur during DSB repair in mammalian cells. Displacement of histones H2B and H3 was apparent during DSB repair around DSBs that were generated by introduction of the I-PpoI and HO homing enzymes into mammalian cells (*Goldstein et al., 2013*; *Sunavala-Dossabhoy and De Benedetti, 2009*; *Berkovich et al., 2007*). Using these systems, the displacement of H2B from the periphery of nucleosomes around a DSB in mammalian cells was dependent on NBS1, a component of the MRE11-RAD50-NBS1 DNA end resection complex (*Berkovich et al., 2007*). In agreement with the idea that DNA end resection drives nucleosome disassembly, loss of H3, which resides in the central part of the nucleosome and is indicative of disassembly of the entire nucleosome, was only observed in cycling cells (*Goldstein et al., 2013*). In cells arrested in G1 phase, which can only repair DSBs by NHEJ, only H2A-H2B were removed during DSB repair not H3-H4. This suggested that only partial, not complete nucleosome disassembly, accompanies NHEJ in mammalian cells.

The mechanism by which chromatin is disassembled and reassembled during DSB repair is unclear. In yeast, the H3-H4 histone chaperone Asf1 is required for chromatin assembly after HR repair, but this function is due to the role of Asf1 in promoting acetylation of histone H3 on lysine 56 (*Chen et al., 2008*). Nucleolin, a nucleolar histone chaperone for H2A-H2B, was shown to be involved in nucleosome disruption in the vicinity of a DSB within the rDNA locus in human cells (*Goldstein et al., 2013*). The factors involved in chromatin disassembly from DSBs in other regions of the mammalian genome are unknown. Also, how chromatin is reassembled in mammalian cells after DSB repair is unknown.

Insights that may be relevant to chromatin disassembly and reassembly during DSB may be provided by the chromatin dynamics during transcription, replication and UV damage repair. The histone chaperone CAF-1 deposits newly-synthesized H3/H4 onto the DNA following DNA replication and nucleotide excision repair (NER) in vitro and in cells (*Gaillard et al., 1996*; *Smith and Stillman, 1991*). Another histone chaperone, HIRA, deposits the histone variant H3.3/H4 onto the genome at times other than DNA synthesis, particularly during transcription (*Ray-Gallet et al., 2002*). The histone chaperone ASF1 donates the histones to CAF-1 and HIRA, where mammalian ASF1A predominantly delivers the H3.3/H4 to HIRA for replication-independent chromatin assembly and ASF1B predominantly delivers H3.1/H4 to CAF-1 for replication-dependent chromatin assembly (*Tagami et al., 2004*; *Tyler et al., 2001*; *Gurard-Levin et al., 2014*). During NER and prior to the assembly of H3.1 into chromatin, HIRA assembles H3.3 into the chromatin around UV lesions. This was unexpected because repair of UV damage involves DNA synthesis which usually engages CAF-1-mediated H3.1 assembly (*Adam et al., 2013*). This transient deposition of H3.3 primes the chromatin for later recovery of transcription after repair of UV lesions by NER (*Adam et al., 2013*). Whether H3.3 is also incorporated prior to H3.1 onto the DNA during DSB repair is unknown, and the underlying mechanisms that regulate such chromatin alterations would also need to be uncovered.

In this study, we set out to delineate nucleosome dynamics during DSB repair in human cells. We found that DSBs lead to complete nucleosome disassembly around the DNA ends, restricted to

within 750 bp from DNA ends. Unexpectedly the chromatin disassembly is independent of DNA end resection because the repair occurred during NHEJ and during MRE11 inhibition. ATM and INO80 facilitate histone disassembly during DSB repair. Meanwhile, chromatin reassembly after repair required both the HIRA-mediated replication-independent pathway and the CAF-1-mediated replication-dependent pathway, suggesting that they occur in a coordinated manner to reestablish the chromatin structure after DSB repair.

## Results

### Histone disassembly and reassembly around DSBs in human cells

To investigate the dynamics of nucleosomes during DSB repair in mammalian cells, we deployed an I-PpoI homing endonuclease inducible system, as previously described (*Pankotai et al., 2012*) (*Figure 1A*). Addition of 4-hydroxytamoxifen (4-OHT) induces nuclear localization of I-PpoI that was fused to the estrogen receptor nuclear localization domain. At 3 hr after addition of tamoxifen to HEK-293 cells to stimulate nuclear localization, the I-PpoI site within the *SLCO5A1* gene was about 50% cleaved in the cell population, as determined by PCR over the I-PpoI site using flanking primers (*Figure 1A,B*). The DNA level at the I-PpoI site was restored back to over 95% at 6–8 hr, indicating that most of the DSBs had been repaired. The kinetics and efficiency of DNA cutting was confirmed by southern blotting analysis (*Figure 1C*). The fairly synchronous cutting and rapid repair was similar to that observed previously with the same I-PpoI system (*Pankotai et al., 2012*).

Complete local nucleosome disassembly and reassembly occurred during DSB repair, as indicated by ChIP analysis of H3 around the I-PpoI break sites within the *SLCO5A1* gene (*Figure 1D*). The H3 occupancy closely followed the kinetics of the cutting and repair using PCR products centered 200bp to the 5' and 3' of the I-PpoI site (*Figure 1D and 1B*). All data was normalized to histone occupancy at the *GAPDH* gene at each time point, which is not near any I-PpoI sites. Furthermore, it appears that nucleosomes adjacent to the break were disassembled completely from around all the DSBs in the population, because histone occupancy dropped to 50% at the three-hour time point at which 50% of the DNA was cleaved. Histone occupancy returned to the pre-damage levels at the 6–8 hr time point, which is the same time point at which the DNA lesion is approximately 100% repaired, indicating that chromatin reassembly occurs concomitant with DSB repair (*Figure 1D* and 1B). The histone H3 disassembly around the DSB progressively diminished at 500bp and 750bp from the break as compared to 200bp from the I-PpoI site (*Figure 1D*), while no conspicuous histone H3 loss was observed at either 1000bp (*Figure 1D*) or 3000bp (*Figure 1—figure supplement 1A*) from the break. This corresponds to a region of approximately 8 nucleosomes being disassembled during DSB repair in human cells. Importantly the observed histone H3 disassembly around a DSB does not appear to be a consequence of persistent DNA end resection, because the DNA level (ChIP input) around the I-PpoI-induced DNA breaks was not significantly affected by induction of DSBs (*Figure 1D*).

This situation is not unique to the *SLCO5A1* gene, because we observed similar kinetics of DSB induction and repair for I-PpoI sites within the *RYR2* gene (*Figure 1E*, top panel) and the *INTS4* gene (*Figure 1—figure supplement 1B*). These data show that the I-PpoI system generates site-specific DSBs in human cells efficiently that are effectively repaired somewhat synchronously. Furthermore, these data show that chromatin disassembly and reassembly accompany cutting and repair of I-PpoI sites within the human genome.

### Chromatin disassembly and reassembly occur during NHEJ, independent of DNA end-resection

To determine whether the observed nucleosome disassembly and reassembly around DSBs occurred during HR or NHEJ, we knocked-down essential components of each repair pathway. Knockdown of RAD51, the recombinase in HR, had no detectable effect on the kinetics of DSB cutting and repair compared to knockdown with a scrambled RNA, indicating that efficient repair of the I-PpoI site occurs in the absence of HR (*Figure 2A*). By contrast, knockdown of KU80, a key factor in NHEJ, led to the more rapid accumulation of breaks and no detectable repair (*Figure 2A*). This result indicates that NHEJ is the major pathway involved in the repair of this I-PpoI site in asynchronous cells, and is consistent with the failure to observe a DNA repair defect upon RAD51 knockdown. The NHEJ

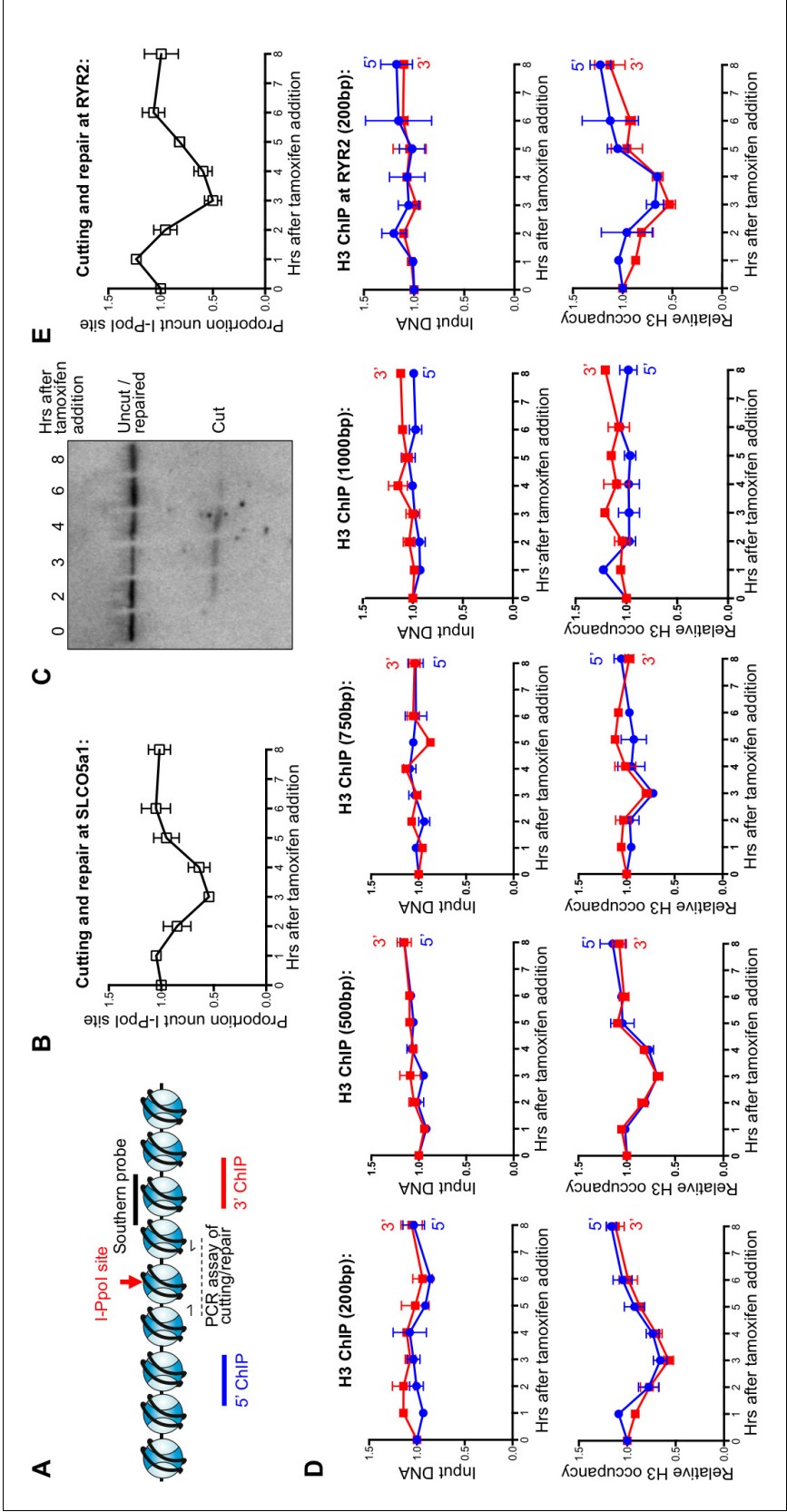

**Figure 1.** Local nucleosome disassembly and reassembly around DSBs in human cells. (**A**) Schematic demonstrating primer pairs spanning the I-PpoI site that were used to measure cutting and repair by quantitative

*Figure 1 continued on next page*

*Figure 1 continued*

PCR, while ChIP analysis was performed on the chromatin to the 5' and 3' of the I-PpoI site. (**B**) The kinetics of generation and repair of the DSB induced in the *SLCO5A1* gene by the I-PpoI endonuclease. Shown is the average +/- SEM from three independent experiments. (**C**) Southern analysis of the kinetics of generation and repair of the DSB induced in the *SLCO5A1* gene by the I-PpoI endonuclease from an independent time course from that shown in (**B**). (**D**) Quantitation of proportion of input DNA and H3 occupancy from ChIP analysis at 200, 500, 750 and 1000 bps (from left to right panels) away from the I-PpoI site within the *SLCO5A1* gene. Shown is the average +/- SEM from three independent experiments. (**E**) (from top to bottom) DNA cutting analysis at the I-PpoI site within the *RYR2* gene, input and H3 occupancy from H3 ChIP analysis 200bp from the I-PpoI site within the *RYR2* gene. Shown is the average +/- SEM from three independent experiments.

The following figure supplement is available for figure 1:

**Figure supplement 1.** Limited H3 disassembly at a distance and I-PpoI cleavage at INTS4.

---

pathway is the preferential DSB repair pathway used in G1 phase of the cell cycle. That repair of the I-PpoI break requires KU80 but not RAD51 is not a consequence of effects of the knockdowns on the cell cycle because flow cytometry analysis detected no apparent change in the cell cycle phase distribution (*Figure 2—figure supplement 1*). As additional evidence that chromatin disassembly and reassembly around the DSB was occurring during NHEJ, we observed efficient chromatin disassembly and reassembly during RAD51 knockdown (*Figure 2A*). Furthermore, chromatin disassembly still occurs in the KU80 knockdown (*Figure 2A*, *Figure 2—figure supplement 2*), indicating that chromatin disassembly occurs prior to recruitment of the core NHEJ machinery.

The MRN complex plays a critical role in detecting, signing and protecting DSBs (*Wyman and Kanaar, 2006*). The nuclease activity of MRE11 regulates the initiation of DNA end-resection and licensing HR (*Shibata et al., 2014*). To investigate if DNA end-resection is required for histone loss, we used two inhibitors of MRE11, PFM39 (a MRE11 exonuclease inhibitor) and PFM03 (a MRE11 endonuclease inhibitor). Addition of either inhibitor had no effect on histone H3 disassembly and reassembly during DSB repair (*Figure 2B*). This is in agreement with the result above that RAD51 knockdown does not disrupt either DNA repair or local chromatin disassembly. Taken together these results indicate that DNA end resection and HR are not required for histone H3 disassembly around DSBs in human cells. Given that local chromatin disassembly around DSBs is unlikely to be driven by the DNA resection machinery physically plowing through nucleosomes, nor requires KU80 recruitment, we sought to identify the trigger for nucleosome disassembly at DSBs.

## ATM, but not ATR, promotes nucleosome disassembly around DSBs

ATM (ataxia-telangiectasia-mutated) and ATR (ATM- and Rad3-related) are the central sensor kinases in the DNA damage checkpoint. Whereas ATM is primarily activated by blunt-ended DSBs, activation of ATR is mostly triggered by ssDNA lesions (*Rhind, 2009*). To determine whether activation of the DNA damage checkpoint via ATM or ATR signaling initiates chromatin assembly, we applied ATM (KU55933) and ATR (VE-821) inhibitors to the I-PpoI repair assay. Inhibition of ATR did not disrupt the histone H3 disassembly and reassembly around the DSB (*Figure 3A*). Upon ATM inhibition, DNA repair was less efficient, as would be expected given the many roles of the DNA damage checkpoint in promoting DNA repair events (*Figure 3B*). Upon ATM inhibition, chromatin disassembly was delayed by about 2 hr as compared to the DNA cutting, where the time point of maximal DNA cutting (3 hrs) showed no significant chromatin disassembly (*Figure 3B*). In addition, the maximal loss of histones was less pronounced around the DSB upon ATM inhibition. Specifically, histone occupancy dropped to only about 75% at the time points where DNA was 50% cut during ATM inhibition (*Figure 3B*), in comparison to the drop of histone occupancy to about 50% at the time points where DNA was 50% cut in the absence of inhibitor. As such, ATM promotes timely and efficient nucleosome disassembly around a DSB.

To determine whether ATR may have a role in triggering chromatin disassembly that is only revealed in the absence of ATM, we inhibited both ATM and ATR (*Figure 3—figure supplement 1*). The results of inhibition of both ATM and ATR were not significantly different from those obtained from ATM inhibition alone. Given that ATR is activated by single-strand DNA, these results provide

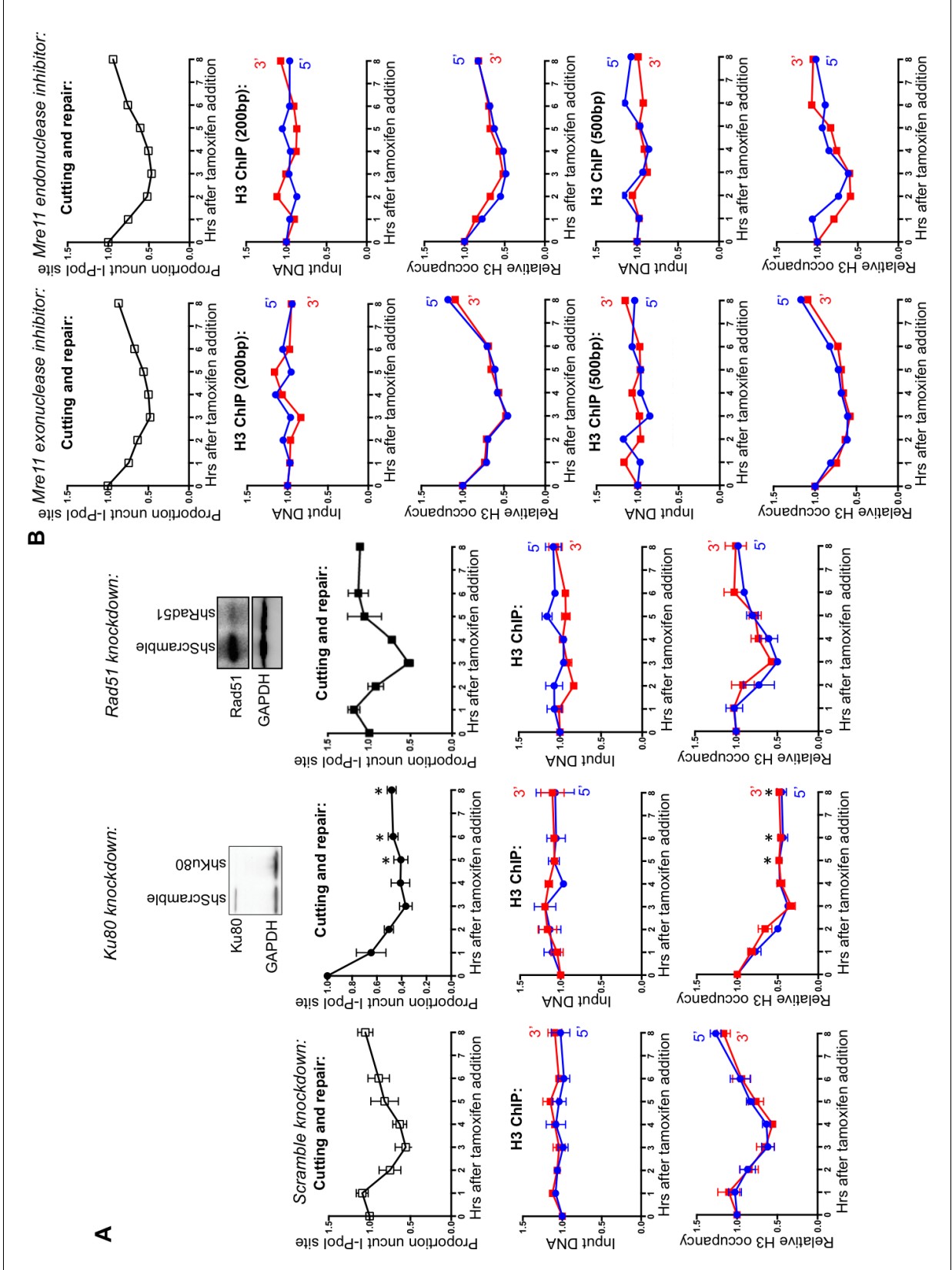

**Figure 2.** DNA resection is not required for histone H3 disassembly. (**A**) The repair of the I-PpoI sites is by NHEJ not HR. At the top is shown western-blotting of the RAD51 and KU80 knockdowns; below is shown the kinetics of generation and repair of the DSBs at *SLCO5A1* I-PpoI sites in scramble

*Figure 2 continued on next page*

*Figure 2 continued*

shRNA knockdown (left), sh-hKU80 (center) and sh-hRAD51 (right) knockdown cells respectively. At the bottom is shown quantitation of the proportion of input DNA and H3 occupancy from ChIP analysis at 200 bps from the I-PpoI site within the *SLCO5A1* gene in the scrambled shRNA knockdown (left panels), the KU80 knockdown (center panels) and RAD51 knockdown (right panels) knockdown cells. Shown is the average +/- SEM from three independent experiments. Asterisks indicate significant changes from the scramble shRNA knockdown, p = 0.005, as determined by the Student's t-test. (B) Inhibition of MRE11 has no effect on repair or chromatin disassembly / reassembly around the I-PpoI site. The panels on the left are experiments performed with cells grown with the MRE11 exonuclease inhibitor PFM39, while those on the right were from experiments performed with cells grown with the MRE11 endonuclease inhibitor PFM03. At the top is shown the kinetics of generation and repair of the DBSs at *SLCO5A1* I-PpoI sites; below is the quantitation of proportion of input DNA and H3 occupancy from ChIP analysis at 200 and 500 bps from the I-PpoI site within the *SLCO5A1* gene.

The following figure supplements are available for figure 2:

**Figure supplement 1.** Flow cytometry analysis of propidium iodide stained DNA, in cells with the indicated knockdowns.

**Figure supplement 2.** H3 disassembly in the absence of KU80.

further evidence that DNA end resection is not involved in chromatin disassembly around a DSB in human cells.

## INO80, but not SMARCAD1, promotes nucleosome disassembly around DSBs in human cells

Nucleosome disassembly is mediated by histone chaperones in concert with ATP-dependent nucleosome remodelers. All our efforts to identify histone chaperones involved in chromatin disassembly during DSB repair to date have failed to find the responsible histone chaperones (data not shown). Therefore, we turned our attention to the ATP-dependent remodelers, especially the ones that are known to be phosphorylated by ATM in response to DSBs. Budding yeast Fun30 promotes DNA end resection and chromatin disassembly around DSBs (*Costelloe et al., 2012*; *Chen et al., 2012*). Its mammalian counterpart SMARCAD1 promotes end resection, presumably in a similar way (*Rowbotham et al., 2011*) (*Costelloe et al., 2012*; *Chen et al., 2012*). Therefore, we examined whether SMARCAD1 also drives chromatin disassembly around DNA break ends in mammalian cells. Consistent with SMARCAD1 playing a role in chromatin disassembly, there was a pronounced delay in the chromatin disassembly around the DSB, upon SMARCAD1 knockdown (*Figure 4A* lower panel) as compared to scrambled RNA knockdown (*Figure 4A*, and data not shown). However, the kinetics of induction of the DSB was also similarly delayed (*Figure 4A* top panel), indicating that the delayed chromatin disassembly was a consequence of the delayed DNA cutting and that SMARCAD1 played no role in promoting chromatin disassembly in the inducible DSB system that we were using. We sought to discover whether SMARCAD1 led to a chromatin structure that was less accessible to the I-PpoI endonuclease and found no evidence for this (data not shown). However, we found that the induction of the I-PpoI endonuclease itself was markedly reduced upon SMARCAD1 knockdown (*Figure 4B*), which was presumably due to a role of SMARCAD1 directly or indirectly in its expression. These results highlight the need to be careful not to over interpret the role of ATP-dependent chromatin remodelers, which are often involved in many genomic processes.

The remodeling complex INO80 (Inositol requiring 80) was previously shown to bind DSBs and assist DSB repair in yeast (*Morrison and Shen, 2009*). In addition, INO80 is phosphorylated by ATM in response to DSBs in yeast (*Morrison et al., 2007*). Therefore, we tested whether INO80 was involved in nucleosome disassembly around a DSB in human cells. Upon INO80 knockdown, the DNA breaks accumulated faster than normal (*Figure 4C*), which is characteristic of conditions in which DSB repair is blocked, such as the KU80 knockdown (*Figure 2A*). Indeed, DNA repair did not occur upon INO80 knockdown (*Figure 4C*). Compared to DNA cutting, chromatin disassembly was markedly delayed upon INO80 knockdown. These results indicate that the ATP-dependent chromatin remodeler INO80 is required for timely chromatin disassembly around a DSB in human cells.

Given that ATM phosphorylates INO80 (*Morrison et al., 2007*), we were interested to see if ATM and INO80 had an epistatic relationship during chromatin disassembly. Consistent with this prediction, we found that combined inhibition of ATM in cells with IN080 knockdown, led to the same

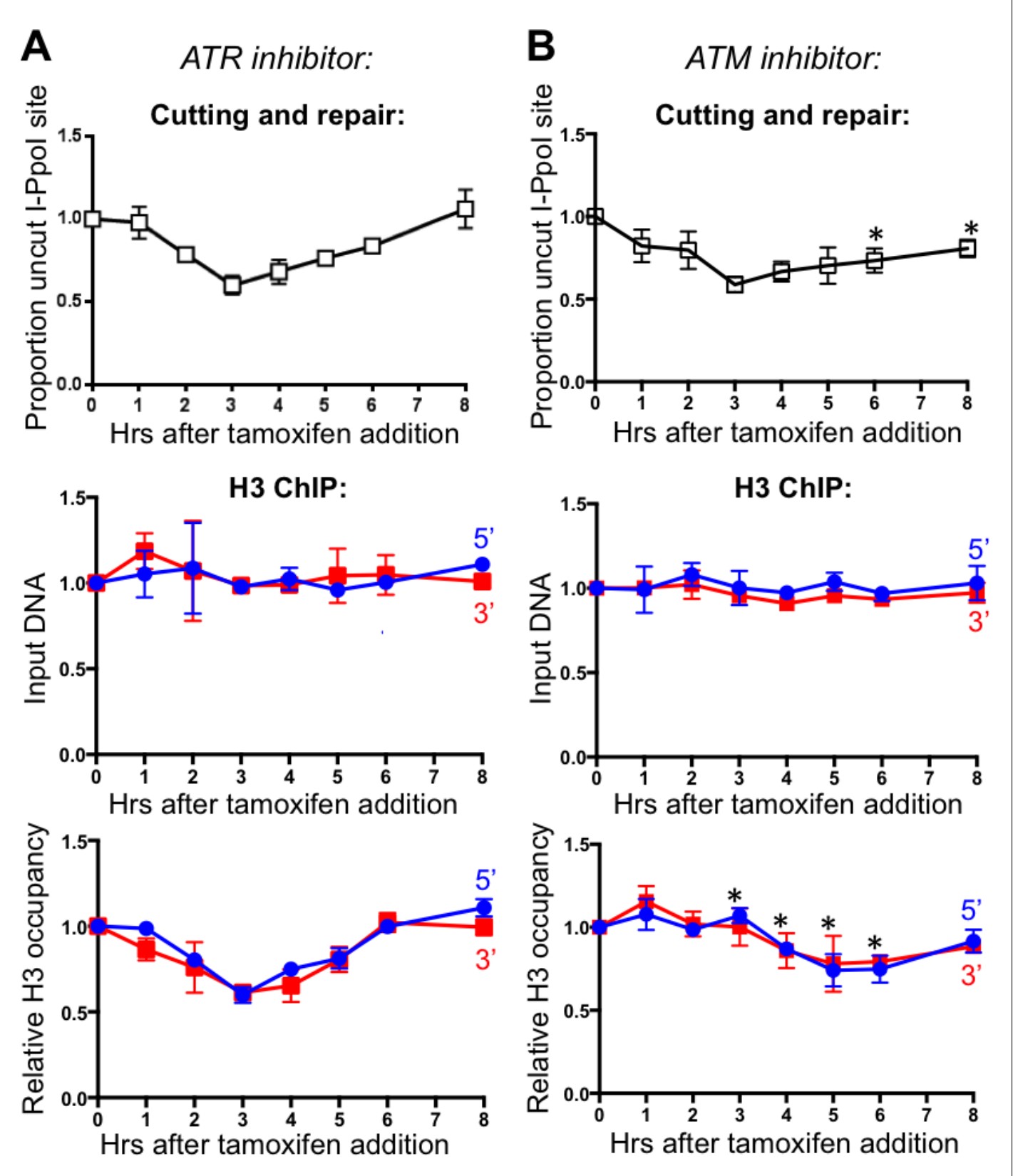

**Figure 3.** ATM, but not ATR, promotes histone H3 disassembly. (**A**) The kinetics of generation and repair of the DBSs at *SLCO5A1* I-PpoI sites is shown at the top; and quantitation of proportion of input DNA and H3 occupancy from ChIP analysis at 200 bps from the I-PpoI site within the *SLCO5A1* gene

*Figure 3 continued on next page*

*Figure 3 continued*

in cells treated with ATR inhibitor VE-821 is shown below. Shown is the average +/- SEM from three independent experiments. (B) As in A, but with treatment with ATR inhibitor KU55933. Asterisks indicate significant changes from no inhibitor, p=0.005, as determined by the Student's t-test.

The following figure supplement is available for figure 3:

**Figure supplement 1.** Effect of combined ATM and ATR inhibitors on chromatin disassembly.

degree of defect in chromatin disassembly as single INO80 knockdown (*Figure 4—figure supplement 1*).

## ASF1A, CAF-1 and HIRA promote chromatin assembly after DSB repair

Our previous work in yeast indicated that the histone chaperone ASF1 is required for chromatin assembly after DSB repair via HR (*Chen et al., 2008*). We knocked down the mammalian paralogs ASF1A and ASF1B to determine whether chromatin reassembly after NHEJ in human cells also required ASF1. ASF1A knockdown led to a complete block in chromatin reassembly after DSB repair (*Figure 5A*, compare lower panel to top panel). Meanwhile, knockdown of ASF1B led to a subtle delay in chromatin reassembly compared to the kinetics of DSB repair (*Figure 5B*). ASF1A/B functions together with the HIRA chaperone in the assembly of the replication-independent variant of H3, H3.3, while ASF1A/B also functions with CAF-1 in the assembly of replication-dependent histone H3.1 and H3.2. To determine which chromatin assembly pathway functions after DSB repair in humans, we knocked down CAF-1 and HIRA. Knockdown of CAF-1 led to a major defect in chromatin reassembly after DSB repair (*Figure 5C*), suggesting that the CAF-1-dependent chromatin assembly pathway is utilized during DSB repair in human cells, even in the absence of significant DNA resection or DNA synthesis. These results were obtained with two independent shRNAs. We observed that knockdown of HIRA led to a complete block in chromatin reassembly after DSB repair with two independent shRNAs (*Figure 5D*), demonstrating that replication-independent chromatin assembly is also important for restoring the chromatin structure after DSB repair in human cells. Because there is very little chromatin assembly after DSB repair in the absence of CAF-1 and virtually no chromatin assembly after DSB repair in the absence of HIRA, it would appear that the replication-dependent and replication-independent chromatin assembly pathways are interdependent for restoring the chromatin structure following DSB repair (*Figure 6*).

## Discussion

Here we have shown that nucleosome disassembly and reassembly around DSBs is a highly regulated and coordinated part of the cellular response to DSB damage in mammalian cells, even during repair pathways that do not involve DNA resection. Repair of DSBs by NHEJ was accompanied by the apparent complete disassembly of nucleosomes adjacent to the break, spanning up to 1500 bp around the break. We found that chromatin disassembly around the DSB is a checkpoint-stimulated response to breaks involving an ATP-dependent chromatin remodeler. Meanwhile, HIRA-mediated assembly of H3.3 and CAF-1 mediated assembly of H3.1 occur after DSB repair in an interdependent manner, suggesting that coordinated mechanisms exist to accurately reestablish the epigenetic information carried by the chromatin structure after DNA repair (*Figure 6*).

## Chromatin disassembly and reassembly during NHEJ in mammalian cells

In agreement with our findings, previous work has shown that histone H3 loss occurs around a DSB in asynchronous human cells (*Goldstein et al., 2013*). However, in that study histone H3 loss was not detected during G1 phase when NHEJ is the predominant form of repair for the non-nucleolar regions of the genome (*Goldstein et al., 2013*). The authors attributed the observed H3 loss around a DSB within the rDNA locus in asynchronous cells to be accompanying HR (*Goldstein et al., 2013*). Notably, their H3-H4 loss reached over 7 kb away from the break ends with no clear decrease at this distance from the break, suggesting it could be a lot more extensive (*Goldstein et al., 2013*). By contrast, in our analyses, histone H3 was disassembled during NHEJ and the histone H3 disassembly was a local event, restricted to within 0.75 kb away from DSBs. One explanation for the difference in

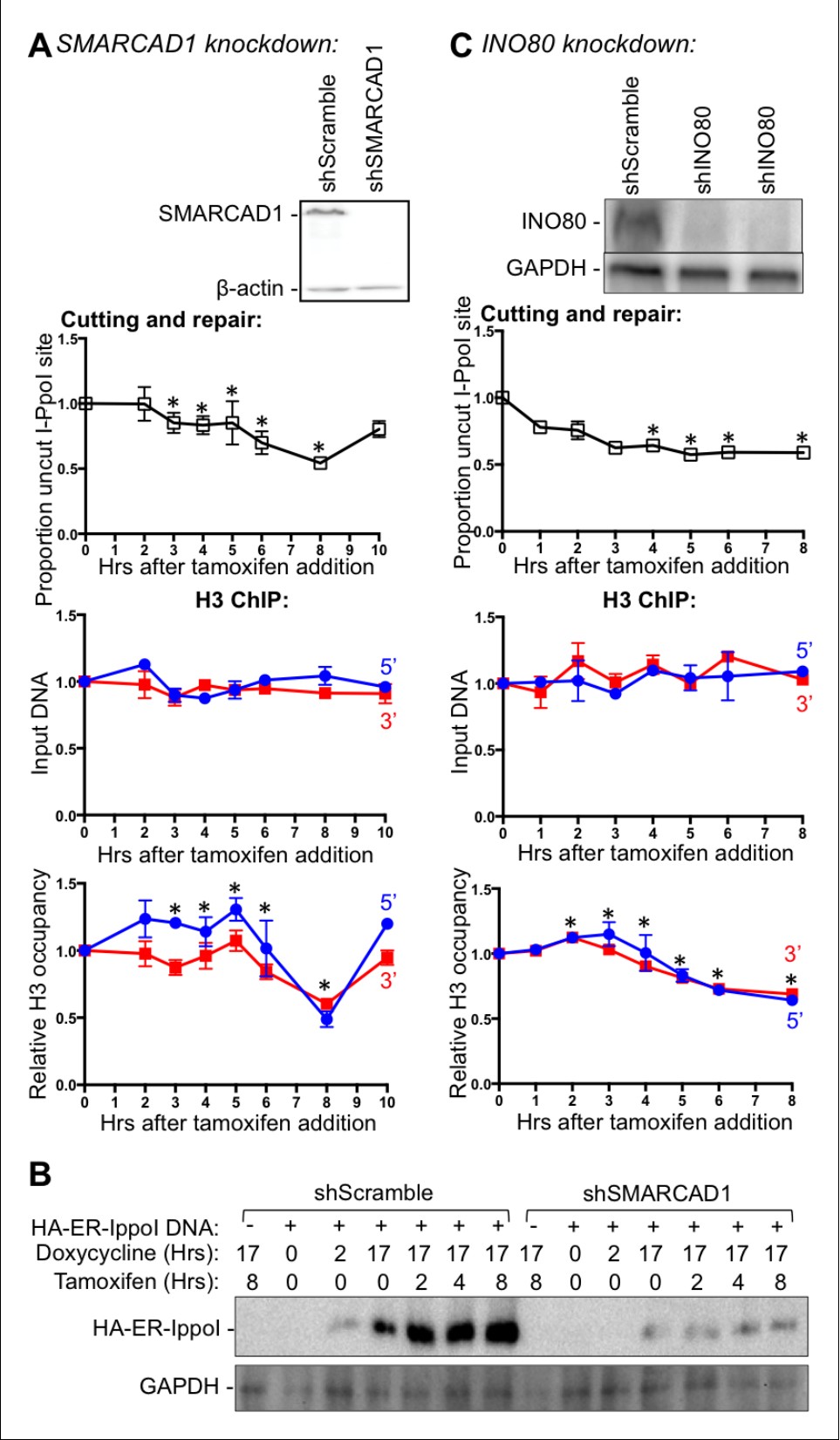

**Figure 4.** INO80, but not SMARCAD1, regulates histone H3 disassembly. (**A**) Western-blotting of SMARCAD1 knockdown is shown at the top. The kinetics of generation and repair of the DBSs at *SLCO5A1* I-PpoI sites is

*Figure 4 continued*

shown below; and quantitation of proportion of input DNA and H3 occupancy from ChIP analysis at 200 bps from the I-PpoI site within the *SLCO5A1* gene in SMRACAD1 knockdown cells is shown at the bottom. Shown is the average +/- SEM from three independent experiments. Efficient cutting and repair was seen in control knockdown cells performed in parallel (data not shown). Asterisks indicate significant changes from the scramble shRNA knockdown, p = 0.005, as determined by the Student's t-test. (B) Kinetics of I-PpoI expression in Scrambled shRNA treated and SMARCAD1 knockdown cells respectively. (C) As in A, but for INO80 knockdown.

The following figure supplement is available for figure 4:

**Figure supplement 1.** Effect of combined ATM inhibition and INO80 knockdown on chromatin disassembly is no more severe that INO80 knockdown alone.

results could be due to the fact that the previous analysis of chromatin disassembly examined I-PpoI sites within the rDNA in the nucleolus. In humans, the rDNA is present in approximately 300 copies of tandem repeats and has a specialized chromatin structure, generally being heterochromatic. Furthermore, the rDNA has a unique nucleolar DNA damage response, enabling HR-mediated repair of I-PpoI-induced DSBs within the rDNA at any phase of the cell cycle, including G1, presumably templated by rDNA repeats in *cis* (*van Sluis and McStay, 2015*). As such, the chromatin disassembly study from the Kastan group was likely examining chromatin disassembly at the rDNA during homologous recombination (*Goldstein et al., 2013*), while we examined chromatin disassembly from euchromatin during NHEJ, which could explain the difference in size of the domain of chromatin disassembly. The failure to observe complete nucleosome disassembly around a DSB within the rDNA in G1 phase by Kastan's group may reflect the unique chromatin structure of the rDNA repeats and their region-specific DNA damage response (*van Sluis and McStay, 2015*). While we have not completely ruled out all types of end-resection in our study, many lines of evidence indicate that the I-PpoI sites we examined are being repaired by an NHEJ mechanism that does not involve much end resection including: (i) the failure to see loss of the DNA signal around the break in the ChIP input, (ii) the lack of involvement of ATR, (iii) the lack of need for the nuclease activities of MRE11 for repair, (iv) no requirement for the single-stranded binding protein RAD51 for repair, (v) the complete dependency on KU80 for DNA repair, and (vi) no requirement for SMARCAD1 in chromatin disassembly, which only plays a role during DNA resection (*Rowbotham et al., 2011*) (*Costelloe et al., 2012*; *Chen et al., 2012*). As such, all the available evidence points to the chromatin disassembly and reassembly around the DSBs induced within the euchromatic genes that we examined, occurring during NHEJ repair.

## Mechanism and function of chromatin disassembly during NHEJ

Given that the molecular events during NHEJ repair occur at the DNA ends, it was somewhat unexpected to find complete disassembly of up to 8 nucleosomes around a DSB during NHEJ. One possibility is that the local histone disassembly may facilitate efficient recruitment of the NHEJ repair machinery to DSBs, while avoiding the loss of epigenetic information that would occur due to extensive histone disassembly. In agreement with a role for chromatin disassembly in efficient NHEJ, knockdown of INO80 or inhibition of ATM, both of which compromise chromatin disassembly around a break, lead to DNA repair defects (*Figures 3 and 4*), although this could also be due to additional roles of ATM and INO80 in the DNA damage response. INO80 has been shown previously to be recruited to DSBs in mammalian cells (*Kashiwaba et al., 2010*), although its exact function there was not clear. It was interesting to find a role for INO80 in promoting chromatin disassembly around a DSB, because in yeast, the role of INO80 in chromatin disassembly around a DSB is dependent on INO80's role in promoting DNA resection (*Tsukuda et al., 2005*; *van Attikum et al., 2004*). However, in our studies in mammalian cells, INO80 promotes chromatin disassembly in the absence of apparent DNA resection, suggesting that it can directly modify chromatin structure during DSB repair. In agreement, INO80 is required to promote accessibility of the NER machinery to UV lesions in mammalian cells (*Jiang et al., 2010*). Given that ATM phosphorylates INO80 (*Morrison et al., 2007*), it will be interesting to determine whether the involvement of ATM in stimulating chromatin disassembly during NHEJ is mediated via it phosphorylating, and potentially activating, INO80.

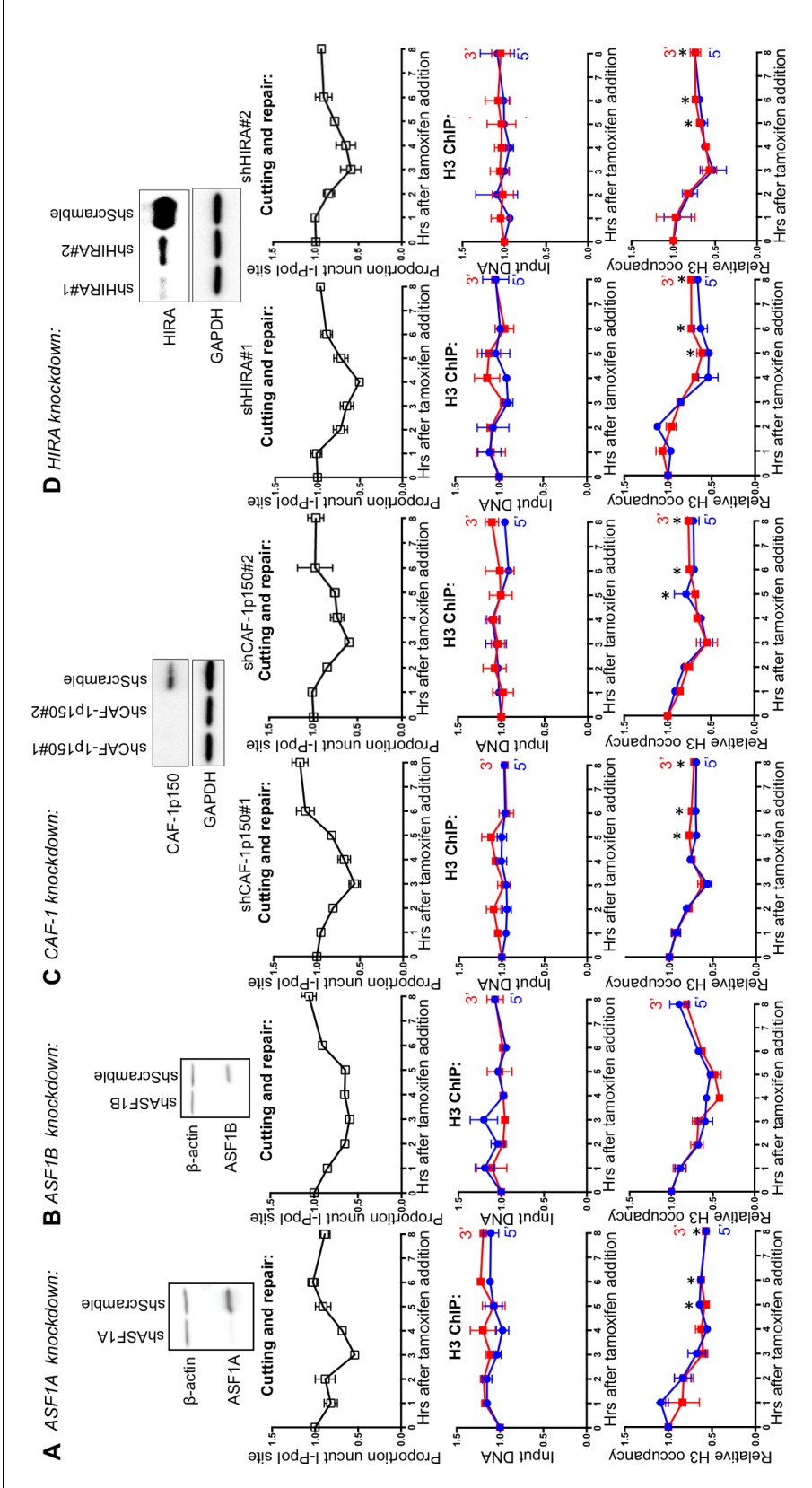

**Figure 5.** ASF1A, CAF-1 and HIRA promote chromatin reassembly. (**A**) Western-blotting of ASF1A knockdown is shown at the top. The kinetics of generation and repair of the DBSs at the *SLCO5A1* I-PpoI site is shown below; *Figure 5 continued on next page*

*Figure 5 continued*

and quantitation of proportion of input DNA and H3 occupancy from ChIP analysis 200 bps from the I-PpoI site within the *SLCO5A1* gene in ASF1A knockdown cells is shown at the bottom. Shown is the average +/- SEM from three independent experiments. Asterisks indicate significant changes from the scramble shRNA knockdown, p=0.005, as determined by the Student's t-test. (**B**) As in **A**, but for ASF1B. (**C**) As in A, but for CAF-1. Data on the left and right are from two independent shRNAs as indicated. (**D**) As in C, but for HIRA.

Consistent with this prediction, we found that combined inhibition of ATM in cells with IN080 knockdown, led to the same degree of defect in chromatin disassembly as INO80 knockdown alone. The function of INO80 in chromatin disassembly at a DSB presumably depends on its histone eviction activity and its ATPase activity, although further study is needed to characterize the detailed molecular mechanism. It is plausible that histone disassembly during DSB repair is mediated by the collaboration between a variety of nucleosome remodelers and chaperones, as well as other factors besides INO80. On the other hand, different nucleosome remodelers may play distinct, but non-redundant roles in chromatin disassembly during DSB repair, as we showed that SMARCAD1 is dispensable for this function (*Figure 4A*).

## Mechanism and function of chromatin reassembly after NHEJ

Chromatin reassembly occurs after DSB repair and is not required for DSB repair per se (*Figure 5*). We found that the histone chaperones ASF1A and ASF1B play non-redundant roles in histone H3 reassembly during DSB repair, with ASF1A playing an essential role in regulating histone reassembly around a DSB, while ASF1B is dispensable for this function (*Figure 5*). ASF1A/B donates histones to the H3.3 HIRA histone chaperone, and in agreement, HIRA was required for chromatin assembly after DSB repair. ASF1A/B also donates histones H3.1 and H3.2 to CAF-1, and CAF-1 knockdown also led to a major defect in chromatin reassembly after DSB repair (*Figure 5C*). In agreement, both HIRA and CAF-1 are recruited to sites of NER, and HIRA deposits H3.3 in response to UV irradiation (*Adam et al., 2013*). However, this previous study was not able to quantitate chromatin assembly during NER repair due to the random nature of the UV induced DNA damage. Given that we were inducing DSBs at known locations, we were able to measure the local histone occupancy. We saw a complete defect in chromatin assembly when we knocked down HIRA. The involvement of HIRA is consistent with the repair being via NHEJ, which does not involve significant DNA synthesis. Indeed, HIRA has been shown to mediate a gap-filling mechanism of H3.3 deposition without the need for transcription or any sequence specificity, given the non-specific DNA binding ability of HIRA (*Ray-Gallet et al., 2011*).

Somewhat unexpectedly, we observed an almost complete block in chromatin assembly after DSB repair when we knocked down CAF-1 (noteworthy, the CAF-1 knockdown was not complete, so its contribution could be even stronger in reality), suggesting that CAF-1 is required for chromatin assembly after DSB repair without ongoing DNA synthesis. It is unlikely that these results are due to indirect effects of CAF-1 knockdown, for example on cell cycle progression, because the defect in chromatin reassembly was localized specifically to the region around the DSB. Furthermore, all the cells would not have had a chance to progress through S phase within the short time course of cutting and repair that occurred in our experiments. The role of CAF-1 in chromatin assembly after DSB repair is not due to an indirect effect on HIRA levels, because no change in HIRA protein level was seen upon knockdown of CAF-1. In agreement with our findings, a recent study found specific recruitment of both the Hir complex and CAF-1 to meiotic DSBs in yeast (*Brachet et al., 2015*). This recruitment of CAF-1 to DSBs during meiosis was not dependent on PCNA nor on DNA strand invasion, indicating that CAF-1 recruitment to DSBs was independent of DNA synthesis per se (*Brachet et al., 2015*). Furthermore, both H3.3 and H3.1 have been observed to be recruited to sites of DSB damage, in laser microirradiation analyses (*Luijsterburg et al., 2016*). These results, together with ours, suggest that the replication-dependent and replication-independent chromatin assembly pathways function in an interdependent manner after DSB repair. Such a concerted mechanism may be necessary to allow faithful reassembly of chromatin structures after DSB repair, perhaps in order to enable the accurate re-establishment of transcription programs. Future studies will reveal how these two pathways are coordinated at the molecular level. It will also be important to

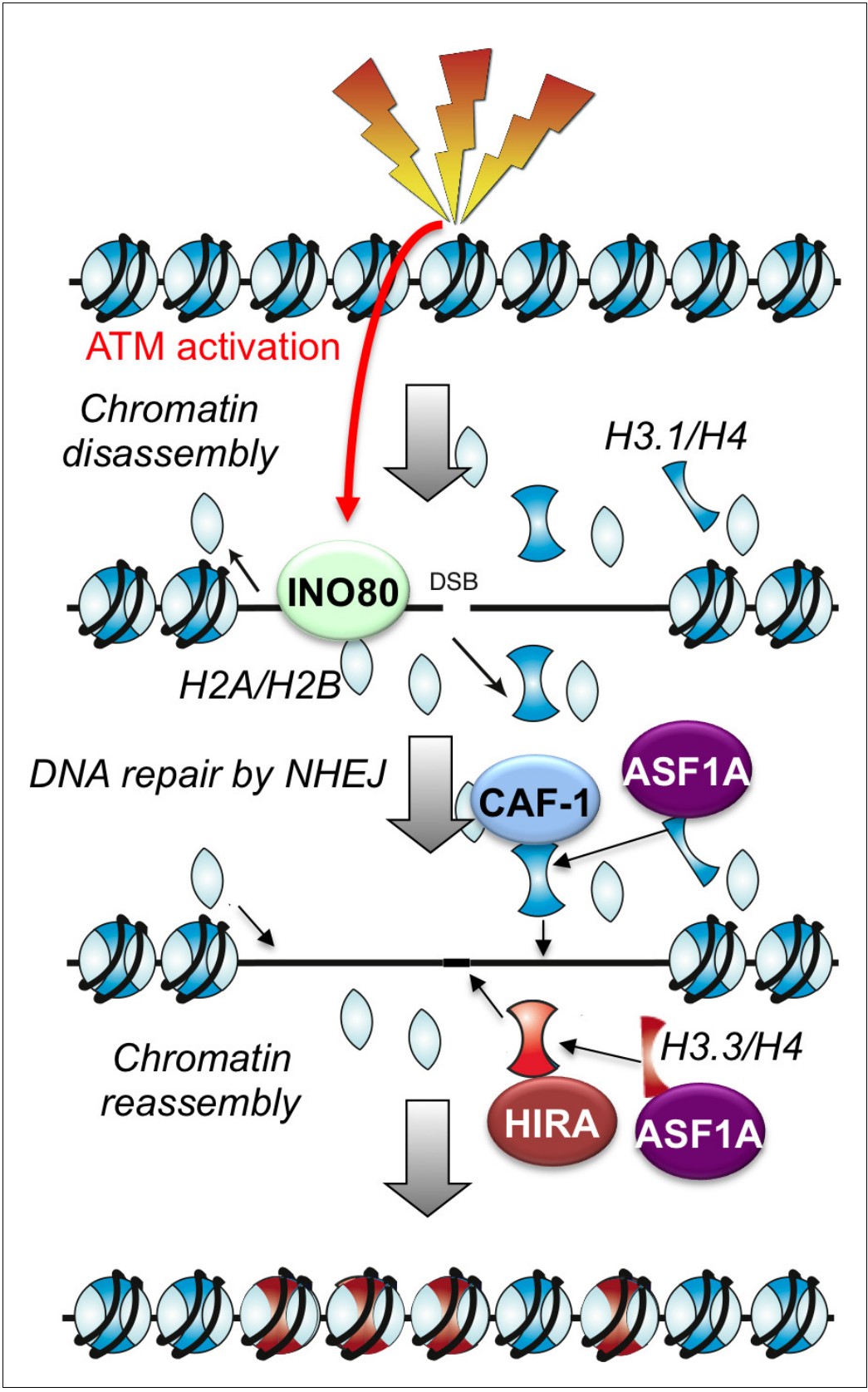

**Figure 6.** Schematic illustration of working model.

determine whether the pattern of H3 variant deposition after DSB repair is the same as that which existed prior to the DNA damaging event.

In conclusion, we propose that immediately following the induction of DSBs in mammalian cells, histone disassembly occurs on both sides of the break. This step is tightly regulated as only about 4 nucleosomes are disrupted on each side of the DNA break, presumably for efficient recruitment of the NHEJ repair machinery. ATM and histone remodelers such as INO80 actively participate in the histone disassembly. After DSB repair, histones are reassembled back to the repaired DNA regions to restore their local chromatin structures, and this step requires functional coordination between histone chaperones ASF1A, CAF-1 and HIRA.

## Materials and methods

### DNA repair assays

HEK293T cell lines were maintained in DMEM containing 10% FBS. The cell lines were obtained from ATCC and were tested regularly for mycoplasma contamination. The procedure for induction of DSBs by I-PpoI was as described (*Pankotai et al., 2012*). Briefly, 3 µg of plasmid expressing I-PpoI fused to the ER nuclear localization domain was transfected into 30 million cells using lipofect-amine-2000. After 16 hr, a final concentration of 1 µM of doxycycline was added to induce expression of I-PpoI, and 12 hr later, 4 µM of tamoxifen was added. The ATM inhibitor Ku55933 (Selleckchem) and ATR inhibitor VE-821 (Selleckchem) were used at 10 µM final concentration, and they were added 1 hr prior to the addition of 4-Hydroxytamoxifen (4-OHT, Sigma). PFM03 (MRE11 endo-inhibitor) and PFM39 (MRE11 exo-inhibitor) (generous gifts from Dr. John Tainer) were used at 100 nM final concentration, and they were added 2 hr prior the addition of 4-OHT (*Shibata et al., 2014*). DNA repair was assayed by real time PCR using the primers (listed below), and the data obtained were normalized to the PCR signal obtained at the GAPDH gene, which is distant from all I-PpoI sites. All experiments were performed independently three times and the average data from three independent experiments are shown +/- SEM.

### shRNA knockdown

GIPZ Lentiviral shRNA constructs (Thermo scientific) corresponding for each target gene were used to generate stable knockdown cells. Production of lentivirus and transfection of HEK293T cell were carried out according to the manufacturer's instructions (Thermo scientific). The efficiency of knockdown for each target gene was confirmed by western blotting. The antibodies used were: RAD51 (abcam), KU70 (cell signaling), SMARCAD1 (abcam), HIRA (active motif), INO80 (abcam), ASF1A (abcam), ASF1B (abcam), CAF1-p150 (abcam), hGAPDH (cell signaling), β-actin (abcam).

### Chromatin immunoprecipitation

Histone H3 Chromatin immunoprecipitation (ChIP) was performed as previously described (*Berkovich et al., 2007*). In addition, ChIP protocol was adapted for use with magnetic bead (Dynabeads® Protein A) for increased efficiency and specificity of immunoprecipitation. The antibody used for histone H3 ChIP was ab1791 (Abcam, Cambridge, MA). The information for primer sequences used in real-time PCR is listed below. Relative occupancy was calculated based on the Cycle threshold (Ct) values. The H3 occupancy and input of the I-PpoI sites were normalized to that at GAPDH at each time point which is distant from any I-PpoI sites and showed no change in histone occupancy and all data were subsequently normalized to the H3 occupancy at time 0 as 1. ChIP data are shown as mean ± SEM of three independent experiments.

### Southern and western analyses

Extracted genomic DNA from HEK293T cell was digested with PstI-EcoRV. DNA samples were resolved by a 1% agarose gel, and transferred to a nylon membrane. The uncut and cut *SLCO5a1* fragments were detected using a radiolabelled *SLCO5a1* probe (primer sequences used for amplifying this probe are listed below).

Western analysis of I-PpoI was performed using an antibody against HA-tag (BioLegend, San Diego, CA, USA).

**Primers used for ChIP, PCR and southern probes:**

| Primer name | Sequence |
| --- | --- |
| SLCO5a1F-Cut | CCCAGTGCTCTGAATGTCAA |
| SLCO5a1R-Cut | CCATTCATGCGCGTCACTA |
| SLCO5a1F-Right (200 bp) | CAAACCATTCATCTCCTTGCATC |
| SLCO5a1R-Right (200 bp) | CATTCACATCGCGTCAACAC |
| SLCO5a1F-Left (200 bp) | GCATGAATGGATGAACGAGAT |
| SLCO5a1R-Left (200 bp) | CAAGCTCAACAGGGTCTTCT |
| SLCO5a1F-Right (500 bp) | GGGTTTGTGACTTGAGCTACT |
| SLCO5a1R-Right (500 bp) | AATCATCCTGTCCAACAGTCTC |
| SLCO5a1F-Left (500 bp) | ACTACAAGCCTACAGTAACCAAA |
| SLCO5a1R-Left (500 bp) | AGTTGTAGATATGCGGCGTTAT |
| SLCO5a1F-Right (750 bp) | CCATTGTGTAATGTGAGTTATTGC |
| SLCO5a1R-Right (750 bp) | GTACTAAGTGATTCATCAGCACAT |
| SLCO5a1F-Left (750 bp) | TGAAACTGGATCCCTTCCTTAC |
| SLCO5a1R-Left (750 bp) | CCCATGGCTATGTCCTGAAT |
| SLCO5a1F-Right (1000 bp) | AGCTCTAGGGATCTGGTGTT |
| SLCO5a1R-Right (1000 bp) | CGGTGGTTAGAATACCTAACATGA |
| SLCO5a1F-Left (1000 bp) | AGAGCTTCTGCACAGCAAA |
| SLCO5a1R-Left (1000 bp) | ATTAGCCCTTTGTCAGATGAGT |
| SLCO5a1F-Right (3000 bp) | AGTCACCTAAGTGCCTGAGA |
| SLCO5a1R-Right (3000 bp) | ACCACTAATAACACATCAGGTTCA |
| SLCO5a1F-Left (3000 bp) | GTCTTACTCTCCTTCGTGTAGTC |
| SLCO5a1R-Left (3000 bp) | GTTTCAGGCCTGCCATTTAG |
| RYR2F-Cut | GTGCTCTGAATGTCAAAGTGAAG |
| RYR2R-Cut | AGGTAGGGACAGTGGGAAT |
| RYR2F-Right (200 bp) | TAGAGCCATGGAAGTCAGAATG |
| RYR2R-Right (200 bp) | CACTGGGCAGAAATCACATC |
| RYR2F-Left (200 bp) | GATGAACGAGATTCCCACTGTC |
| RYR2R-Left (200 bp) | AGCTCAACAGGGTCTTCTTTC |
| DAB1F-Cut | CTCCATTTCAGAGCTGAGCA |
| DAB1R-Cut | GCTGAGATCCCCAAGATTCA |
| DAB1F-Right (200 bp) | CCAACTCCTTCACCAGCA |
| DAB1R-Right (200 bp) | GCTCAGCTCTGAAATGGAGAT |
| DAB1F-Left (200 bp) | CATACCACAGTGGAAAGAGCA |
| DAB1R-Left (200 bp) | GGGCATAGGAAGGTGAAGTTTA |
| INTS4F-Cut | GACAGTGGGAATCTCGTTCAT |
| INTS4R-Cut | ATTCAATGAAGCACGGGTAAAC |
| Probe-SLCO5A1F | AGCTCTAGGGATCTGGTGTT |
| Probe-SLCO5A1R | GTACTAAGTGATTCATCAGCACAT |
| GAPDH-F | TCAGCCAGTCCCAGCCCAAG |
| GAPDH-R | GAGAAAGTAGGGCCCGGCTAC |

## Acknowledgements

We are indebted to Evi Soutoglou for kindly providing I-PpoI expression vectors and troubleshooting advice. We are grateful to Barry Sleckman and Caitlin Purman for input and advice on Southern blotting, and John Tainer for generously providing MRE11 inhibitors: PFM03 and PFM39. This work was supported by NIH grant CA95641 (to JKT).

## Additional information

### Competing interests

JKT: Senior editor, eLife. The other author declares that no competing interests exist.

### Funding

| Funder | Grant reference number | Author |
|---|---|---|
| National Institutes of Health | CA95641 | Jessica K Tyler |

The funders had no role in study design, data collection and interpretation, or the decision to submit the work for publication.

### Author contributions

XL, Design and acquisition of data, Analysis and drafting manscript, Conception and design, Acquisition of data, Analysis and interpretation of data, Drafting or revising the article; JKT, Conception and design, Interpretation of data, Revising article, Conception and design, Analysis and interpretation of data, Drafting or revising the article

### Author ORCIDs

Jessica K Tyler, http://orcid.org/0000-0001-9765-1659

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
