## [Decision Letter]

Thank you for submitting your article "Nucleosome disassembly during NHEJ in human cells, followed by concerted HIRA- and CAF-1-dependent chromatin reassembly" for consideration by *eLife*. Your article has been favorably evaluated by Kevin Struhl (Senior editor) and three reviewers, one of whom, Jerry Workman, is a member of our Board of Reviewing Editors.

The reviewers have discussed the reviews with one another and the Reviewing Editor has drafted this decision to help you prepare a revised submission.

Summary:

This is nice paper that assists in resolving the critical factors that contribute to double strand break (DSB) repair. In particular, the requirements for chromatin remodeling and histone chaperones are demonstrated in reconstructing the chromatin environment during induction of DSBs and repair. The authors utilize an inducible system to create site-specific DSB in mammalian cells. Through multiple lines of evidence (chemical inhibitors and measurements of input DNA) DNA end resection is not evident during repair or nucleosome (de)assembly. Nevertheless, nucleosome eviction is still observed and found to be dependent on ATM and the INO80 chromatin remodeler. In the absence of this activity, repair is deficient. Histone chaperones are then needed to reassemble the chromatin surrounding DNA breaks. The logic of the manuscript is concise and the data largely convincing.

Essential revisions:

1) A major concern is whether the apparent loss of histones next to the break might be an artifact of the break itself. If the ChIPed DNA fragments are sheared to on average 500 base pairs then a ChIP signal could be seen from histones bound on either side of the primers within 500 bps. However, if you cut next to the primers with the targeted nuclease then the ChIP signal can only come from histones bound within 500 base pairs of the other side (i.e. bound on one side but not both sides). Hence an artificial apparent drop in histone cross linking. The authors need to test or convincingly deal with this possibility.

2) At times it is difficult to discern significant differences in H3 occupancy after DSB formation between wildtype and mutant cell lines. p-values should be calculated when authors suggest significant alterations.

3) Not all of the panels have error bars. The nature of the assay is not robust, measuring less than 2 fold changes. Thus the statistics are important and error bars are needed.

4) The connection between ATM and INO80 function in nucleosome disassembly is not particularly strong. Are the authors suggesting that ATM regulates INO80 to facilitate nucleosome disassembly? If so can a more direct assay be performed, such as shINO80 knockdown in the presence and absence of ATM inhibitor to assess epistatic or added effects.

5) Most experiments in this manuscript used only one shRNA to knock down target protein. The authors should include at least one more shRNA to exclude off-target effects. This is an important issue as the effect of knockdown of some genes on nucleosome disassembly assembly is small (maximum 2 fold). Alternatively, it is a good idea to reintroduce the protein to see whether the defects can be rescued by the effects.

6) In Figure 4, the authors showed that depletion of SMARCAD1 reduces the expression level of endonuclease IppoI. Therefore, it is important to analyze the IppoI expression level in every experiment to make sure it is not affected, especially in INO80 mutant cells.

7) I suspect that nucleosome disassembly/assembly and repair efficiency depends on the cell cycle. The effect of INO80 knockdown. KU80 and Rad51 on cell cycle should be analyzed.

8) It is hard to imagine that knockdown CAF-1 and HIRA affect nucleosome assembly similar after DNA repair. If this is the case, one would expect to detect H3.1 and H3.3 at break sites similarly.

---

## [Author Response]

*Essential revisions:*

1) A major concern is whether the apparent loss of histones next to the break might be an artifact of the break itself. If the ChIPed DNA fragments are sheared to on average 500 base pairs then a ChIP signal could be seen from histones bound on either side of the primers within 500 bps. However, if you cut next to the primers with the targeted nuclease then the ChIP signal can only come from histones bound within 500 base pairs of the other side (i.e. bound on one side but not both sides). Hence an artificial apparent drop in histone cross linking. The authors need to test or convincingly deal with this possibility.

It is true that introducing a DSB could lead to a decrease of the potential DNA fragments that can be ChIPed down. However, this is not the reason for the observed histone level decrease for the following reasons. The sonicated fragment size is generally about 500 bps in our assays, while the closest primer set from the break end is centered around 200bps away. Given that our replicon size is about 100-120bps, this means that the closest primer end in our assay is still about 140-150 bps away from break end. Even if that statistical counting indeed has an effect, the effect should be no more that 25%. In addition, we also detect 50% decrease at 500 bps away, as well as at 750 away. These two sites should not be affected by that statistical model. Moreover, we show that certain conditions (knockdown of INO80 and ATM inhibition) lead to no histone removal at specific time points that have efficient cutting. If the apparent histone loss is an artifact of the break itself, these results would have never have been obtained. In addition, the fact that knockdown of the histone chaperones ASF1, CAF-1 and HIRA all have effective DNA repair, yet there is no return of the histones by ChIP, shows that this signal is not an artifact of the break itself reducing the ChIPability of the flanking DNA. If it were, the ChIPed signal would have returned in the ASF1, CAF-1 and HIRA knockdowns upon DNA repair. The approach and interpretation of histone ChIP around a break that we use is the norm in the field, and is also used by the Soutoglou and Kastan labs.

2) At times it is difficult to discern significant differences in H3 occupancy after DSB formation between wildtype and mutant cell lines. p-values should be calculated when authors suggest significant alterations.

p values have now been calculated for the knockdown cell lines and inhibitors, and significantly different time points from the control cells are indicated.

3) Not all of the panels have error bars. The nature of the assay is not robust, measuring less than 2 fold changes. Thus the statistics are important and error bars are needed.

The only data that did not previously have error bars was the experiments using the ATR and MRE11 inhibitors, both of which showed no change from the wild type / no drug control. As such, we were not making statements of novel roles for these proteins from these particular data.In addition, all experiments included the analysis of two DNA regions of equal distance to the 5’ and 3’ of the IPpoI site, that showed highly consistent results, strikingly so for the MRE11 inhibitor experiments. In the case of the MRE11 inhibitor experiments, we also performed the ChIP at both 200bp and 500bp away at both 5’ and 3 ‘to the IPpoI site and saw the identical result. The analysis of four different regions itself indicates the reproducibility of the effect and could be considered more meaningful than analyzing the same region multiple times. But as requested by the reviewers, we have now repeated the ATR inhibitor analyses two more times to provide error bars. Unfortunately, we were not able to do this for the MRE11 inhibitor experiments, given that they were provided in limited quantities by a colleague from my previous institution, so were not readily available, in addition to the time limitations and large amounts of other additional experiments that were needed for this revision.

4) The connection between ATM and INO80 function in nucleosome disassembly is not particularly strong. Are the authors suggesting that ATM regulates INO80 to facilitate nucleosome disassembly? If so can a more direct assay be performed, such as shINO80 knockdown in the presence and absence of ATM inhibitor to assess epistatic or added effects.

As requested, we have now combined INO80 knockdown with ATM inhibition, and found that there is no synergistic increase. This is consistent with ATM modifying INO80 to promote chromatin disassembly, but far from proves it. We have included the new data and discussed it in the text accordingly.

5) Most experiments in this manuscript used only one shRNA to knock down target protein. The authors should include at least one more shRNA to exclude off-target effects. This is an important issue as the effect of knockdown of some genes on nucleosome disassembly assembly is small (maximum 2 fold). Alternatively, it is a good idea to reintroduce the protein to see whether the defects can be rescued by the effects.

These are good points, and we have now repeated the critical findings of the paper with additional shRNAs, to show that the results are not off target effects. The data are now included in the revised manuscript.

6) In Figure 4, the authors showed that depletion of SMARCAD1 reduces the expression level of endonuclease IppoI. Therefore, it is important to analyze the IppoI expression level in every experiment to make sure it is not affected, especially in INO80 mutant cells.

In Figure 4 we showed that upon knockdown of SMARCAD1, I-PpoI mediated cleavage of the DNA occurred at around 8 hrs, as opposed to 3hrs in the normal situation. This is what led us to analyze the IppoI induction and show that they are indeed greatly delayed and reduced upon SMARCAD1 knockdown. However, in the other mutants, the kinetics and extent of I-PpoI mediated cleavage of the DNA is not significantly altered and as such the functional read out of I-PpoI activity shows that I-PpoI is being expressed. Western blots of I-PpoI for every experiment would provide no additional information.

7) I suspect that nucleosome disassembly/assembly and repair efficiency depends on the cell cycle. The effect of INO80 knockdown. KU80 and Rad51 on cell cycle should be analyzed.

Our data show that in asynchronous cells, the repair pathway that is used to repair the I-PpoI sites that we examine is NHEJ. NHEJ occurs efficiently throughout the cell cycle, but it is likely that there are differences in nucleosome disassembly/assembly pathways used through the cell cycle. Accordingly, as requested, we have examined the effects on INO80, KU80 and Rad51 knockdown on the cell cycle and find no significant alteration in the proportion of cells in G1, S, or G2/M. This data is now included as Figure 2—figure supplement 1.

*8) It is hard to imagine that knockdown CAF-1 and HIRA affect nucleosome assembly similar after DNA repair. If this is the case, one would expect to detect H3.1 and H3.3 at break sites similarly.*

In agreement with our findings, a paper a few months ago from Haico van Attikum’s group (Luijsterburg et al., Mol Cell, 2016) indeed shows incorporation of H3.1 and H3.3 at sites of double strand breaks, using SNAP labeling of newly-synthesized histones. We have now cited this paper.